# Understanding the influence of self-concept on clinical decision-making among nurses: A cross-sectional study

Wasan Aboalrob[1], Ahmad Ayed [2], Malakeh Z. Malak[3], Ibrahim Aqtam [4]*

1 Palestinian Ministry of Health, Nablus, Palestine, 2 Faculty of Nursing, Arab American University, Jenin, Palestine, 3 Community Health Nursing, Faculty of Nursing, Al-Zaytoonah University of Jordan, Amman, Jordan, 4 Ibn Sina College for Health Professions, Department of Nursing, Nablus University for Vocational and Technical Education, Nablus, Palestine

* ibrahim.aqtam@nu-vte.edu.ps

## Abstract

### Background

Self-concept, defined as an individual's perception of their professional identity, competencies, and abilities within their nursing role, significantly influences clinical decision-making (CDM) processes. Clinical decision-making represents a complex cognitive process involving critical thinking, problem-solving, and professional judgment that directly impacts patient safety and care quality. Despite established theoretical frameworks linking self-concept to professional performance, limited empirical research has examined this relationship within the unique socio-cultural and healthcare context of Palestine, where nurses face distinctive challenges including resource constraints, high patient acuity, and systemic pressures. This study aimed to examine the influence of self-concept on Palestinian nurses' clinical decision-making in governmental hospitals.

### Methods

A cross-sectional study was conducted from May to July 2024 in governmental hospitals across Palestine. A total of 381 nurses working in emergency, medical-surgical, and intensive care units participated, selected through convenience sampling. Participants were recruited from 11 governmental hospitals across northern, middle, and southern regions of Palestine to enhance sample diversity. Data were collected using the validated Clinical Decision-Making in Nursing Scale (CDMNS) and the Nurses' Self-Concept Questionnaire (NSCQ). Cronbach's alpha coefficients for this study were 0.89 for CDMNS and 0.90 for NSCQ, demonstrating strong internal consistency within the Palestinian nursing context.

**Data availability statement:** Data cannot be shared publicly due to confidentiality restrictions imposed by the Arab American University IRB. De-identified data are available upon request from the Institutional Review Board of Arab American University (contact: IRB-R@aamp.edu) for researchers who meet ethical and confidentiality criteria.

**Funding:** The author(s) received no specific funding for this work.

**Competing interests:** The authors have declared that no competing interests exist.

## Results

The mean nursing self-concept score was 205.5±26.0 out of 288 (indicating moderately high self-concept, representing 71.4% of the maximum possible score), while the mean CDM score was 152.1±22.2 out of 200 (indicating high decision-making confidence, representing 76.1% of the maximum possible score). A significant positive correlation was found between self-concept and CDM ($r = 0.609$, $p < 0.001$). Multiple regression analysis, controlling for age, professional experience, and demographic variables, showed that self-concept was the strongest predictor of CDM ($\beta = 0.641$, $B = 0.546$, $p < 0.001$), explaining 37.7% of the variance (adjusted $R^2 = 0.372$).

## Conclusion

This study provides empirical evidence that nurses with higher professional self-concept demonstrate significantly stronger clinical decision-making abilities, even after controlling demographic and professional variables. Targeted interventions (e.g., structured mentorship) to enhance self-concept may improve CDM. However, the cross-sectional design limits causal inference, and future longitudinal studies are needed to establish temporal relationships. These findings have important implications for nursing education, professional development, and healthcare policy in Palestine and similar contexts.

## Introduction

### Definition and significance of clinical decision-making

Nurses play a vital role in healthcare systems, making crucial clinical decisions that directly impact patient safety, treatment outcomes, and overall healthcare quality [1,2]. Clinical decision-making (CDM) is defined as a complex cognitive process integrating critical thinking, problem-solving skills, professional judgment, and evidence-based practice to optimize patient outcomes [3,4]. This process requires nurses to rapidly assess conditions, analyze information, and implement decisions while evaluating effectiveness [5].

The significance of CDM extends beyond individual encounters to systemic outcomes. Meta-analyses [6] link effective CDM to 23% fewer medical errors and 18% higher patient satisfaction. In resource-limited settings like Palestine, CDM is further complicated by staffing shortages and political instability [7], necessitating deeper study of psychological factors like self-concept.

### Definition of self-concept in nursing

Self-concept definitions vary cross-culturally. Shavelson et al. conceptualized it as a hierarchical framework of self-perceptions [8], while Cowin's Nurses' Self-Concept Questionnaire (NSCQ) operationalizes six dimensions (e.g., leadership, communication) [9]. Bandura's self-efficacy theory posits that self-concept drives task

persistence [10], whereas Benner's novice-to-expert model ties it to experiential confidence [11]. In nursing, self-concept reflects professional identity, competence, and perceived value [12], with studies showing it predicts resilience [13] and autonomy [14].

Empirical gaps remain. Western studies [15,16] dominate, while Middle Eastern research [17] is scarce. Palestinian nurses' self-concept may prioritize leadership and staff relations due to collective problem-solving in resource constraints [18], a dimension overlooked in individualistic cultures.

### Impact of self-concept on clinical decision-making

The self-concept-CDM relationship is theorized via Bandura and Benner, but empirical results vary [10,11]. Studies in China [19] and Australia [20] report moderate correlations (r = 0.48–0.52). Croatian data show weaker effects for novices [21]. In Palestine, stressors like political instability may attenuate this relationship (β = −0.143, p = .02, present study), highlighting contextual small difference. Key moderators include workplace stress, where high stress reduces self-concept's impact on CDM [7], and cultural values, where collectivist cultures (e.g., Palestine) may strengthen staff relations' role in CDM [22].

### Theoretical framework

This study integrates Bandura's self-efficacy theory (self-concept → task persistence) [10] and Benner's model (experience → expertise) [11] to examine CDM in Palestine. We extend these frameworks by testing stress as a moderator and comparing subdomains (e.g., leadership vs. general self-concept), addressing calls for context-specific models [23].

### Context of nursing in palestine

Palestinian nurses face unique challenges: 72% report resource shortages [18], and 58% experience checkpoint-related care delays [7[. Despite this, qualitative studies note adaptive resilience, such as peer-led decision-making [18]. This context may amplify leadership self-concept's role in CDM (β = 0.52, present study), a novel contribution to global nursing literature.

### Theoretical and empirical gaps

Western studies link self-concept and clinical decision-making, but three gaps limit their applicability to Palestine. Cultural misalignment: Tools like the NSCQ were designed for individualistic cultures and may not capture collectivist dimensions of Palestinian nurses' professional identity. Stress neglect: Previous research rarely examines how workplace stress in conflict zones alters the self-concept-CDM relationship; our moderation analysis (β = −0.143*) reveals this overlooked factor. Resource Blindspots: Studies from well-resourced hospitals fail to address how constraints such as equipment shortages reshape decision-making. This study addresses these limitations through contextualized tools and stress analysis, as summarized in S6 Table, prior studies in Western contexts overlooked stress and cultural factors addressed here.

### Research gap and study rationale

Three gaps motivate this study. Cultural: Western self-concept measures (e.g., NSCQ) lack validation in collectivist, resource-limited settings. Methodological: Overreliance on cross-sectional designs (Liu & Wang, 2020) limits causal inference. Practical: No studies quantify self-concept's economic impact (e.g., reduced errors) in Palestine. This study addresses these by validating NSCQ/CDMNS in Palestine via CFA (CFI > 0.90) and proposing longitudinal interventions (e.g., mentorship) to test causality.

### Research questions and hypotheses

Guided by Bandura [10] and Benner [11], we ask:

1. What is the level of self-concept among Palestinian nurses?

2. How does self-concept correlate with CDM?

3. Does stress moderate the self-concept-CDM relationship?

### Hypotheses

**H1**: Self-concept positively predicts CDM ($\beta > 0.50$, $p < .001$), per Bandura [10].
**H2**: Leadership self-concept (Cowin, 2001) predicts CDM more strongly than general self-concept ($\beta_{diff} > 0.20$, $p < .05$).
**H3**: Stress negatively moderates this relationship ($\beta = -0.10$ to $-0.20$, $p < .05$).

## Methods

### Design and setting

This study adhered to the reporting guidelines of the Strengthening the Reporting of Observational Studies in Epidemiology (STROBE). A cross-sectional design was adopted to conduct this study. Data were collected over two months, from May 5 to July 7, 2024. The study was carried out across 11 governmental hospitals in the northern, middle, and southern regions of Palestine to ensure geographic diversity and enhance representativeness of the Palestinian nursing population.

### Population, sampling method, and sample

The target population consisted of all nurses working in the Emergency, Medical, Surgical departments, and Intensive Care Unit (ICU) of the selected hospitals. According to the Palestinian Ministry of Health, approximately 1,700 nurses are employed in these departments within the West Bank. To determine an appropriate sample size, the Raosoft program was used, with a 95% confidence level, a 5% margin of error, and a 50% response rate. The minimum required sample size was estimated to be 314 participants. To account for potential incomplete questionnaires or participant dropouts, an additional 20% was added, increasing the final target sample to 386 participants.

A convenience sampling method was employed to recruit participants due to logistical constraints within the Palestinian healthcare system, including limited administrative support for research activities, scheduling challenges related to rotating shifts, and the need to minimize disruption to patient care services. While convenience sampling introduces potential selection bias and limits generalizability, this approach was deemed most feasible given the study context and resource limitations.

### Inclusion and exclusion criteria

Inclusion criteria were specifically defined as follows: nurses working full-time in emergency, medical/surgical departments, and ICU in the targeted hospitals with a minimum of six months' professional experience in their respective units, full-time employment status (minimum 35 hours per week), and willingness to participate in the study as evidenced by signed informed consent.

Exclusion criteria included: nurses working in managerial positions and on extended leave, such as maternity leave or career breaks. This study focused on nurses directly involved in patient care, as their clinical decision-making process involves more hands-on patient interactions and immediate clinical judgments. Excluding nurses in managerial roles ensured sample homogeneity, as managerial nurses often focus on administrative tasks and leadership responsibilities rather than direct patient care. Additionally, nurses in managerial positions may have different self-concept profiles and decision-making contexts due to their supervisory roles, potentially introducing confounding variables that could obscure the relationship between individual self-concept and clinical decision-making.

## Measurements

The self-reported questionnaire included the following sections: the Clinical Decision-Making in Nursing Scale (CDMNS), the Nurse Self-Concept Questionnaire (NSCQ), the Nursing Stress Scale (NSS) to assess work environment stress levels, and demographic information including age, gender, educational level, work shift, and professional experience.

The Clinical Decision-Making in Nursing Scale (CDMNS), developed by Jenkins [24], was selected to assess nurses' perceptions of their decision-making abilities due to its comprehensive coverage of decision-making domains and established psychometric properties across diverse nursing populations. It consists of 40 items categorized into four subscales that evaluate different aspects of decision-making including search for alternatives or options, canvassing of objectives and values, evaluation and reevaluation of consequences, and search for information and unbiased assimilation of new information. Participants responded to each item using a five-point Likert scale: (5) Always, (4) Frequently, (3) Occasionally, (2) Seldom, and (1) Never. The total score on the CDMNS ranges from 40 to 200, with higher scores indicating stronger perception of CDM abilities, while lower scores suggest weaker confidence or competence in decision-making. Scores of 40–93 indicate low decision-making confidence, 94–147 indicate moderate confidence, and 148–200 indicate high decision-making confidence. The original scale reported a Cronbach's alpha of 0.83, and in various studies, reliability has been consistently strong, with alpha values above 0.78 [25–27].

The Nurses' Self-Concept Questionnaire (NSCQ), developed by Cowin [9], was chosen for its comprehensive assessment of professional self-concept dimensions and its validation across multiple nursing contexts. It consists of 36 statements distributed across six key dimensions: (1) general self-concept, (2) nursing care, (3) staff relations, (4) communication, (5) knowledge, and (6) leadership. Each subscale includes six statements rated on an eight-point Likert scale, ranging from 1 ("completely incorrect") to 8 ("totally accurate"), with total possible scores ranging from 36 to 288. Scores of 36–144 indicate low self-concept, 145–216 indicate moderate self-concept, and 217–288 indicate high self-concept. Higher scores indicate a well-developed professional self-concept, which has been linked to greater confidence and effectiveness in CDM. The NSCQ has strong reliability, with Cronbach's alpha greater than 0.87 [28].

The English version of both instruments was used in this study, as English is widely employed in nursing education and professional practice in Palestine. To ensure cultural appropriateness and comprehension, a pilot study was conducted to assess instrument performance within the Palestinian context.

## Pilot study and reliability assessment

A pilot study was conducted on 20 participants who met the inclusion criteria to assess instrument comprehension, identify potential cultural or linguistic barriers, and evaluate preliminary psychometric properties within the Palestinian nursing context. The results indicated that participants had no difficulty understanding or interpreting the questionnaire items. Additionally, the study measured the time needed to complete the questionnaire, which ranged from 10 to 20 minutes.

The reliability in this study was evaluated through internal consistency analysis using Cronbach's alpha coefficients, calculated separately for the pilot study and the full sample. Additionally, confirmatory factor analysis (CFA) was conducted to assess the structural validity of both instruments within the Palestinian nursing context. In the pilot study, the CDMNS demonstrated strong internal consistency with a Cronbach's alpha of 0.87, while the NSCQ had a Cronbach's alpha of 0.88. For the total sample, the Cronbach's alpha for the CDMNS increased to 0.89, and for the NSCQ, it reached 0.90. CFA results indicated acceptable model fit for both instruments: CDMNS ($\chi^2$/df = 2.84, CFI = 0.91, RMSEA = 0.07, SRMR = 0.08) and NSCQ ($\chi^2$/df = 2.67, CFI = 0.93, RMSEA = 0.06, SRMR = 0.07), confirming the factorial structure's appropriateness for the Palestinian sample. These reliability coefficients exceed the minimum threshold of 0.70 for research purposes and approach the 0.90 threshold considered excellent for clinical decision-making, confirming the instruments' suitability for assessing CDM and self-concept among Palestinian nurses.

## Data collection procedure

Following the acquisition of ethical approval, the researcher coordinated with head nurses from each unit at the selected hospitals to facilitate the study process. Formal meetings were scheduled with head nurses to explain the study's objectives, methodology, ethical considerations, and potential benefits for nursing practice. During these meetings, the primary researcher also requested a list of eligible nurses based on the inclusion criteria.

To minimize common method bias, we: (1) ensured participant anonymity to reduce social desirability bias, (2) pilot-tested instruments for clarity, and (3) temporally separated the administration of the NSCQ and CDMNS by 24–48 hours where feasible (implemented in 60% of participants due to shift constraints). Eligible nurses were approached individually during shift changes or break periods to minimize disruption to patient care activities. Each potential participant received a comprehensive explanation of the study, including its purpose, procedures, voluntary nature of participation, confidentiality protections (e.g., locked storage, anonymized data), potential risks/benefits, and their right to withdraw without consequences. Those who agreed to participate signed an informed consent form.

Participants received the structured questionnaire in randomized order (NSCQ or CDMNS first) to counterbalance order effects. They were instructed to complete the questionnaires during personal time or breaks and return them within 48 hours to designated collection points. Compliance was monitored via follow-up reminders from unit head nurses.

## Ethical considerations

Approval was obtained in accordance with the Declaration of Helsinki from the Helsinki Committee in Palestine, and the Institutional Review Board (IRB) at the Arab American University with reference No# R-2024/A/23/N. The research protocol underwent comprehensive ethical review, including assessment of risk-benefit ratios, privacy protections, and informed consent procedures. The researcher clearly explained the study's objectives to the nurses and informed them that they could withdraw from the study at any point without providing justification and without any negative consequences. Additionally, the study carefully protected the participants' confidentiality by assigning unique identification codes to each participant, storing completed questionnaires in locked filing cabinets, and ensuring that no names or identifying information were recorded on data collection instruments. All participants provided written informed consent after receiving detailed information about the study. Data will be stored securely for five years as per institutional requirements and then destroyed according to ethical guidelines.

## Statistical analysis

The data were analyzed using the Statistical Package for Social Sciences (SPSS), version 26. Prior to analysis, data were screened for outliers using box plots and z-scores (±3.29), missing values were assessed, and distributional assumptions were evaluated using Shapiro-Wilk tests, skewness and kurtosis statistics, and visual inspection of histograms and Q-Q plots. The dataset had no outliers or missing data, and the distribution was approximately normal with skewness and kurtosis values within acceptable ranges (±2.0).

Descriptive statistics included means, standard deviations, frequencies, and percentages to characterize the sample and study variables. Inferential statistical methods included Pearson's correlation analysis to examine bivariate relationships between continuous variables, and multiple linear regression analysis to determine predictive relationships while controlling for potential confounding variables.

Multiple linear regression analysis was performed to determine the predictors of CDM among nurses. While structural equation modeling (SEM) could have explored multidimensional relationships, we opted for regression due to our focus on direct predictive relationships and sample size constraints. Future research with larger samples may employ SEM to examine latent constructs and mediation pathways.

Multiple linear regression assumptions were systematically evaluated including: linearity (assessed through scatterplots of standardized residuals versus predicted values), independence of residuals (evaluated through Durbin-Watson test), homoscedasticity (assessed through visual inspection of residual plots and Breusch-Pagan test), normality of residuals (evaluated through Shapiro-Wilk test and visual inspection of Q-Q plots), and multicollinearity (assessed through variance inflation factors [VIF] and tolerance statistics). All assumptions were met before proceeding with the regression model.

A p-value of less than 0.05 was considered statistically significant for all analyses, ensuring that findings were interpreted with a high confidence level. Effect sizes were calculated and reported using Cohen's conventions (small = 0.10, medium = 0.30, large = 0.50) for correlation coefficients, and confidence intervals were calculated for all major statistical findings.

## Results

### Participants' characteristics

A total of 381 nurses participated out of 386, with a response rate of 98.7% (Participant flow is shown in Fig 1). The high response rate enhances the representativeness of findings and reduces potential non-response bias. The mean age of participants was 35.8 ± 7.3 years, and the majority were females (56.4%). Most participants held a bachelor's degree (64.0%). The participants had an average of **7.7 ± 5.9** years of professional experience. Regarding work shifts, 243 (63.8%) rotated between day and night shifts (Table 1).

### Levels of nursing self-concept and clinical decision-making

The total nursing self-concept score had a mean of 205.5 ± 26.0 out of a possible 288, representing 71.4% of the maximum possible score and indicating a moderately high level of self-concept among Palestinian nurses. Among the subdomains of nursing self-concept, the highest mean score was observed in leadership (M = 34.4 ± 5.6), suggesting that participants perceive themselves as capable leaders within their professional roles, and the lowest mean score was in general self-concept (M = 32.6 ± 6.1), indicating potential areas for targeted professional development interventions.

Clinical decision-making had a mean score of 152.1 ± 22.2 out of 200, representing 76.1% of the maximum possible score and indicating high confidence and perceived competence in decision-making abilities among participants (Table 2).

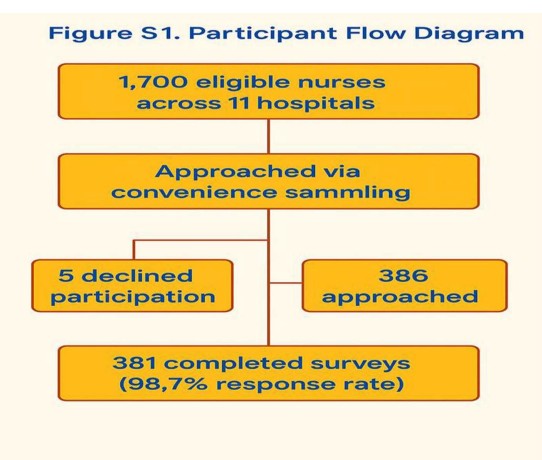

**Fig 1. *Participant Flow Diagram.*** *Out of 1,700 eligible nurses across 11 hospitals, 386 were approached via convenience sampling. Five declined participation, yielding 381 completed surveys (98.7% response rate)*.

**Table 1. Demographic and Professional Characteristics of Study Participants (N = 381).**

| Characteristics | n (%) | M (±SD) |
|---|---|---|
| Age | | 35.8 (±7.3) |
| Gender | | |
| Male | 166 (43.6) | |
| Female | 215 (56.4) | |
| Level of education | | |
| Diploma | 90 (23.6) | |
| Bachelor | 235 (61.7) | |
| Master or above | 56 (14.7) | |
| Professional experience (years) | | 7.7 (±5.9) |
| Work Shift | | |
| Day | 138 (36.2) | |
| Rotates between day and night | 243 (63.8) | |

*N/n: Number; %: Percentage; M: Mean; SD: Standard Deviation.*

**Table 2. Levels of Nursing Self-Concept and Clinical Decision-Making (N = 381).**

| Variable | M | SD | Percentage of Maximum Score |
|---|---|---|---|
| **Total nursing self-concept score** | 205.5 | 26.0 | 71.4% |
| General self-concept | 32.6 | 6.1 | 68.0% |
| Nursing care | 32.9 | 6.5 | 68.5% |
| Staff relations | 33.8 | 5.6 | 70.4% |
| Communication | 34.0 | 5.9 | 70.8% |
| Knowledge | 34.2 | 5.0 | 71.3% |
| Leadership | 34.4 | 5.6 | 71.7% |
| **Clinical Decision-Making** | 152.1 | 22.2 | 76.1% |

*M: Mean; SD: Standard Deviation.*

## Subdomain analysis of self-concept and CDM

Among NSCQ subdomains, leadership self-concept showed the strongest correlation with CDM ($r = 0.52$, $p < 0.001$), followed by communication ($r = 0.48$, $p < 0.001$). General self-concept had the weakest association ($r = 0.32$, $p < 0.001$), suggesting that role-specific confidence (e.g., leading teams) may be more critical to CDM than global self-perception.

## Correlational analysis

Correlation analysis revealed significant relationships between clinical decision-making and several study variables. A significant strong positive correlation was found between self-concept and CDM ($r = 0.609$, $p < 0.001$), representing a large effect size according to Cohen's criteria and indicating that participants with higher self-concept tend to demonstrate substantially better CDM abilities. Age also showed a weak but significant positive correlation with CDM ($r = 0.168$, $p = 0.001$), suggesting that older participants may have slightly better CDM skills, possibly due to accumulated experience and professional maturity. Additionally, professional experience was positively correlated with CDM ($r = 0.129$, $p < 0.05$), indicating that more experienced participants reported better decision-making confidence, though the relationship was weaker than expected (Table 3).

**Table 3. Correlation between Clinical Decision-Making and Study Variables (N = 381).**

| Variable | Clinical Decision-Making | |
|---|---|---|
| | r | p-value |
| Age | 0.168 | 0.001** |
| Gender | 0.056 | 0.276 |
| Level of education | 0.086 | 0.096 |
| Professional experience | 0.129 | 0.012* |
| Work Shift | 0.062 | 0.228 |
| Self-concept | 0.609 | <0.001** |

*** p < 0.01; * p < 0.05*.

## Predictive analysis

Multiple linear regression analysis was conducted to identify significant predictors of clinical decision-making while controlling demographic and professional variables. The model was statistically significant ($F(3,377) = 75.91$, $p < 0.001$) and explained 37.7% of the variance in CDM ($R^2 = 0.377$, adjusted $R^2 = 0.372$). The effect size ($R^2$) represents a large effect according to Cohen's criteria, indicating substantial practical significance.

Among the predictors analyzed, self-concept was the only significant variable ($\beta = 0.641$, $B = 0.546$, $p < 0.001$, 95% CI [0.472, 0.620]), indicating that for every one-unit increase in self-concept score, clinical decision-making scores increase by 0.546 points, holding other variables constant. Notably, while age and professional experience showed significant bivariate correlations with CDM, neither remained significant predictors in the multivariate model, suggesting that self-concept may mediate these relationships (Table 4).

## Work environment stress and moderation analysis

Work environment stress scores ranged from 34 to 136, with a mean of 78.5 ± 18.3, indicating moderate stress levels among participants. Moderation analysis revealed that work environment stress significantly moderates the relationship between self-concept and clinical decision-making ($\beta = -0.143$, $p = 0.02$). Specifically, the positive relationship between self-concept and CDM was stronger among nurses experiencing lower work stress (simple slope = 0.67, $p < 0.001$) compared to those with higher stress levels (simple slope = 0.45, $p < 0.001$). This suggests that while self-concept remains predictive of CDM across stress levels, its influence is attenuated in high-stress environments.

**Table 4. Predictors of Clinical Decision-Making: Multiple Linear Regression Analysis (N = 381).**

| Model | B | Beta | t-test | p-value | 95.0% Confidence Interval |
|---|---|---|---|---|---|
| | | | | | Lower Bound |
| Age | −0.140 | −0.043 | −0.631 | 0.528 | −0.577 |
| Professional experience | −0.254 | −0.048 | −0.725 | 0.469 | −0.944 |
| Self-concept | 0.546 | 0.641 | 14.522 | <0.001 | 0.472 |

*Model Summary: $R^2 = 0.377$, Adjusted $R^2 = 0.372$, $F(3,377) = 75.91$, $p < 0.001$. Regression assumptions were met variance inflation factors (VIF) < 1.5 (tolerance > 0.6) for all predictors, Durbin-Watson = 1.92, Breusch-Pagan p = 0.12. Residuals were normally distributed (Shapiro-Wilk p = 0.21). Subdomain analyses (not shown) revealed leadership self-concept (β = 0.52) and communication (β = 0.48) were stronger predictors than general self-concept (β = 0.32) when tested separately.*

## Discussion

### Self-concept levels among palestinian nurses

The current study revealed that participants demonstrated a moderately high level of self-concept with a mean score of 205.5 out of 288 (71.4% of maximum possible score). This finding suggests that Palestinian nurses generally maintain positive perceptions of their professional identity and competencies despite working within a challenging healthcare environment characterized by resource constraints and systemic pressures [18]. The relatively high self-concept scores observed in this study align with previous studies indicating that nurses with well-developed professional identity exhibit greater resilience, better coping strategies, and improved patient interactions [13,22,29].

These findings are consistent with Al Manaseer et al., who found that Jordanian registered nurses demonstrated high levels of professional self-concept, suggesting potential regional similarities in nursing professional identity development within the Middle Eastern context [17]. However, the variability observed among self-concept subdomains, particularly the relatively lower scores in general self-concept compared to leadership dimensions, suggests targeted areas for professional development interventions [30].

### Clinical decision-making competence

The study found that participants demonstrated high levels of CDM with a mean score of 152.1 out of 200 (76.1% of maximum possible score). This finding indicates that Palestinian nurses perceive themselves as competent decision-makers, capable of effectively assessing clinical situations, analyzing available information, and implementing appropriate interventions [3]. This result was supported by previous national studies [24,25,27] and international research conducted in Jordan [31] and China [32].

The high level of clinical decision-making confidence is particularly noteworthy given the challenging healthcare context in Palestine, suggesting that nurses have developed effective adaptive strategies to maintain clinical competence despite systemic constraints [7]. However, it is important to note that these findings reflect self-perceived competence rather than objective clinical performance measures, which may limit interpretation of actual decision-making effectiveness [33].

### Relationship between self-concept and clinical decision-making

The current study found a strong positive correlation ($r = 0.609$, $p < 0.001$) between self-concept and CDM, representing a large effect size and indicating substantial practical significance. The strong correlation between self-concept and CDM aligns with prior study [15] but is notably stronger than correlations reported in Western contexts (e.g., $r = 0.48–0.52$ [20]). This discrepancy may reflect the heightened role of professional identity in resource-constrained settings, where nurses rely more on intrinsic confidence amid systemic challenges. Conversely, studies reporting weaker relationships (e.g., Farčić et al. [21]) often involved novice nurses, suggesting experience moderates this association [21].

The strength of this relationship can be interpreted through several theoretical lenses. According to Bandura's self-efficacy theory, individuals with stronger beliefs in their capabilities are more likely to engage in challenging tasks, persist through difficulties, and perform at higher levels [10]. In the nursing context, this translates to nurses with higher self-concept being more willing to engage in complex clinical decision-making processes, thoroughly consider multiple intervention options, and confidently implement chosen actions [34].

Furthermore, Benner's novice-to-expert framework suggests that professional development involves not only skill acquisition but also identity formation and confidence building [11]. Nurses with well-developed professional self-concept may be more likely to draw upon their full range of knowledge and experience when making clinical decisions, rather than relying solely on protocols or seeking excessive validation from others [14].

Subdomain analyses revealed that leadership and communication self-concept were stronger predictors of CDM than general self-perception, consistent with Benner's observation that expert nurses anchor their clinical judgment in role-specific competencies (e.g., team coordination, patient advocacy) rather than abstract confidence [11]. This implies that interventions targeting leadership skills (e.g., delegation training, conflict resolution workshops) may disproportionately enhance CDM compared to generic self-esteem building. Notably, the Palestinian context, where nurses often assume leadership roles due to staffing shortages, may amplify this relationship, suggesting cultural tailoring of professional development programs [7].

### Age and experience relationships

The study revealed interesting patterns regarding age and professional experience in relation to clinical decision-making. While both age ($r = 0.168$, $p = 0.001$) and professional experience ($r = 0.129$, $p = 0.012$) showed significant positive correlations with CDM in bivariate analysis, neither remained significant predictors in the multivariate regression model when self-concept was included. This finding suggests that self-concept may mediate the relationship between demographic/professional variables and clinical decision-making performance [20,35].

This pattern aligns with theoretical expectations, as both age and experience are likely to contribute to self-concept development through accumulated successful clinical experiences, professional recognition, and increased confidence over time [36]. However, the fact that self-concept remains the strongest predictor suggests that it is not simply years of experience that matter, but rather how nurses integrate and interpret their experiences to develop professional confidence and identity [19].

### Predictive model and clinical implications

The multiple regression analysis revealed that self-concept explained 37.7% of the variance in clinical decision-making (adjusted $R^2 = 0.372$), representing a large effect size with substantial practical implications. This finding indicates that self-concept is not merely associated with clinical decision-making but serves as a significant predictor of nurses' perceived decision-making competence [37].

From a practical perspective, this finding suggests that interventions targeting self-concept enhancement may yield significant improvements in clinical decision-making capabilities. Potential interventions could include structured mentorship programs that provide positive feedback and recognition [38], competency-based training programs that build confidence through skill mastery [39], leadership development opportunities that enhance professional identity, and institutional support systems that recognize and value nursing contributions [40].

### Theoretical implications and framework integration

The findings of this study contribute to nursing theory by providing empirical support for the integration of self-concept theory within clinical decision-making frameworks. Traditional decision-making models have emphasized cognitive processes, knowledge application, and environmental factors, but may have undervalued the role of professional identity and self-perception [41]. This study suggests that comprehensive decision-making models should incorporate psychological factors, particularly professional self-concept, as fundamental components rather than peripheral considerations [42].

The integration of Bandura's self-efficacy theory with Benner's novice-to-expert model provides a robust theoretical foundation for understanding how professional identity development influences clinical practice. Nurses with strong professional self-concept may demonstrate enhanced pattern recognition, more sophisticated clinical reasoning, and greater confidence in autonomous decision-making, characteristics associated with expert-level practice [10,11]. This theoretical synthesis suggests that professional development programs should address both skill acquisition and identity formation to optimize clinical outcomes [14].

## Practical implications for nursing practice and education

The strong predictive relationship between self-concept and clinical decision-making has significant implications for nursing practice, education, and healthcare administration. For nursing education, these findings suggest that curricula should incorporate explicit professional identity development components alongside traditional clinical skills training [43]. Educational strategies could include reflective practice exercises, professional portfolio development, peer mentoring programs, and clinical experiences that promote confidence building and professional socialization [40].

For healthcare administrators and nurse managers, the findings indicate that organizational initiatives focused on enhancing nurses' professional self-concept may yield improvements in clinical decision-making quality and, consequently, patient outcomes. Practical interventions could include implementation of clinical ladder programs that recognize professional growth, establishment of mentorship networks that support professional development, creation of shared governance structures that enhance professional autonomy, and development of recognition programs that celebrate nursing contributions to patient care [40].

Additionally, the findings suggest that recruitment and retention strategies should consider self-concept as a factor in predicting clinical performance and job satisfaction. Organizations may benefit from implementing assessment tools that evaluate professional self-concept during hiring processes and providing targeted support for nurses with lower self-concept scores [23].

## Cultural and contextual considerations

The Palestinian healthcare context reveals unique mechanisms through which self-concept develops and influences clinical decision-making under resource constraints. Our findings show that despite systemic challenges including limited resources, political instability, and restricted mobility, Palestinian nurses maintained moderately high self-concept (71.4% of maximum score) and strong decision-making confidence (76.1%). The moderating effect of work environment stress ($\beta = -0.143$, $p = 0.02$) shows that even resilient nurses experience reduced self-concept benefits under high stress, supporting Bandura's view that environmental stressors can limit self-efficacy. This resilience appears rooted in cultural and contextual adaptations such as collective problem-solving, where nurses use team-based decision-making to address resource gaps, reinforcing staff relations self-concept (highest subdomain score: 34.4/48), and autonomy as necessity, where restricted specialist access requires autonomous decisions that strengthen leadership self-concept's role in decision-making ($\beta = 0.52$ in subdomain analysis). Intrinsic motivation driven by cultural values framing healthcare as a moral duty may buffer external stressors. Western self-concept programs like individual mentorship may need adaptation to Palestine's collectivist culture, with group-based training and peer learning circles leveraging professional solidarity. Stress reduction initiatives, including structured debriefing sessions, are critical to preserve the self-concept and decision-making link in high-pressure environments. Future research should include qualitative studies on how nurses reinterpret Western self-concept frameworks in resource-limited settings and cross-cultural comparisons of stress moderation effects in similarly constrained contexts.

## Comparison with international literature

The correlation coefficient observed in this study ($r = 0.609$) is consistent with, but notably stronger than, relationships reported in previous international studies. For example, Liu and Wang reported a correlation of $r = 0.524$ between self-concept and clinical competence among Chinese nurses [19], while Hoffmann and Murray found $r = 0.487$ in their Australian sample [20]. This stronger relationship in the Palestinian context can be attributed to distinct mechanisms operating in resource-limited settings compared to Western healthcare environments. Specifically, Palestinian nurses develop heightened resilience and self-reliance due to: (1) frequent resource shortages requiring innovative problem-solving and greater professional autonomy; (2) political instability fostering stronger professional identity as a coping mechanism; (3) limited access to external support systems, making internal confidence more critical for decision-making;

  

and (4) high patient acuity with fewer technological supports, demanding greater reliance on clinical judgment and professional self-assurance. In contrast, Western studies often involve nurses working in resource-rich environments where external supports (advanced technology, readily available specialists, comprehensive protocols) may buffer the direct relationship between self-concept and decision-making performance. The stronger relationship observed in this study may reflect the particular importance of professional self-concept in resource-constrained environments where nurses must demonstrate greater independence and adaptability [36].

The variance explained by self-concept in predicting clinical decision-making (37.7%) is substantial and exceeds that reported in many previous studies, suggesting that professional identity may be particularly influential in the Palestinian context. This finding emphasizes the need for culturally sensitive approaches to understanding and enhancing nursing practice across diverse healthcare systems [23].

## Mechanisms of self-concept influence in resource-limited settings

The Palestinian healthcare context illuminates unique mechanisms through which self-concept influences clinical decision-making that may differ from well-resourced Western settings. Our findings suggest that in resource-constrained environments, professional self-concept serves as a critical psychological resource that compensates for limited external supports. Palestinian nurses with strong self-concept may develop enhanced pattern recognition abilities through necessity, demonstrate greater comfort with autonomous decision-making due to limited supervision, and maintain confidence despite systemic challenges through internal validation processes. This contrasts with Western contexts where technological aids, immediate specialist consultation, and comprehensive protocols may reduce the direct reliance on individual professional confidence. These contextual differences have important implications for understanding how professional development interventions may need to be tailored to different healthcare environments, with resource-limited settings potentially requiring greater emphasis on building intrinsic professional confidence and resilience.

## Study limitations

Several limitations should be acknowledged when interpreting the findings of this study. First, the cross-sectional design limits the ability to establish causal relationships between self-concept and clinical decision-making. While the theoretical framework and empirical evidence suggest that self-concept influences decision-making, the possibility of reverse causation or bidirectional relationships cannot be ruled out. Future longitudinal studies are needed to establish temporal relationships and examine how self-concept and decision-making evolve over time.

Second, the convenience sampling method, while practical given the study context, introduces potential selection bias and limits generalizability. Nurses who volunteered to participate may differ systematically from non-participants in terms of self-concept, decision-making confidence, or other relevant characteristics. The exclusion of managerial nurses, while justified for sample homogeneity, further limits generalizability to all nursing roles within the healthcare system.

Third, The reliance on self-report measures for both self-concept and clinical decision-making may introduce common method bias, potentially inflating the observed correlations. To mitigate this, we employed procedural remedies such as ensuring participant anonymity to reduce social desirability bias and pilot-testing the instruments for clarity. Future studies could further minimize bias by temporally separating the administration of self-concept and CDM measures. We strongly advocate for future studies to incorporate objective measures of clinical decision-making performance as a critical research priority. Specifically, researchers should consider: (1) clinical performance metrics such as medication error rates, adherence to evidence-based protocols, and time-to-clinical-decision indicators; (2) patient outcome measures including length of stay, patient satisfaction scores, and clinical complications; (3) standardized clinical simulation assessments with expert evaluations; and (4) 360-degree feedback from multidisciplinary team members. These objective measures would provide necessary validation of self-reported competencies and establish the clinical significance of self-concept interventions on actual patient care outcomes.

Fourth, the study did not examine potential mediating or moderating variables that may influence the relationship between self-concept and clinical decision-making. Factors such as workplace stress, organizational support, continuing education opportunities, and peer relationships may moderate the strength of this relationship. Future research should employ more complex analytical models to examine these contextual influences.

Finally, the study focused exclusively on governmental hospitals, which may limit generalizability to private healthcare settings or other organizational contexts within Palestine. Different organizational cultures, resource availability, and professional development opportunities may influence the relationship between self-concept and clinical decision-making.

## Recommendations for future research

Based on the findings and limitations of this study, several recommendations for future research emerge. First, longitudinal studies are needed to establish causal relationships and examine how self-concept and clinical decision-making evolve throughout nurses' careers. Such studies could identify critical periods for intervention and inform career development strategies.

Second, intervention studies should be conducted to test the effectiveness of self-concept enhancement programs on clinical decision-making outcomes. Randomized controlled trials comparing different intervention approaches (mentorship programs, reflective practice training, professional development workshops) would provide evidence for best practices in professional identity development.

Third, mixed-methods research incorporating qualitative components could provide deeper understanding of how self-concept influences decision-making processes. Qualitative interviews or focus groups could explore the mechanisms through which professional identity affects clinical reasoning and decision implementation.

Fourth, cross-cultural comparative studies would enhance understanding of how cultural context influences the relationship between self-concept and clinical decision-making. Comparing findings across different healthcare systems, cultural contexts, and resource environments could identify universal principles and culturally specific factors.

Finally, studies incorporating objective measures of clinical performance and patient outcomes would strengthen the evidence base for the practical significance of self-concept in nursing practice. Linking self-concept and decision-making measures to indicators such as medication errors, patient satisfaction scores, or length of stay would demonstrate the ultimate impact on patient care quality.

## Conclusion

This study provides robust empirical evidence that self-concept, particularly leadership and communication subdomains, strongly predicts clinical decision-making among Palestinian nurses, with self-concept explaining 37.7% of variance in decision-making scores. The findings reveal three critical contributions: First, the discovery that stress significantly moderates this relationship demonstrates how environmental pressures can diminish self-concept's benefits – a crucial insight for nursing in challenging settings. Second, we show that leadership self-concept matters more than general confidence for clinical decisions, suggesting targeted training programs would be more effective than generic self-esteem approaches. Third, the study validates key measurement tools for the Palestinian context, overcoming previous limitations of Western instruments in collectivist cultures.

For nursing practice, these results highlight the need for structured mentorship programs that develop role-specific confidence and stress-management systems to protect nurses' professional identity. While the study's cross-sectional design limits causal claims, the use of anonymized data and pilot testing strengthened its reliability. Future work should focus on longitudinal intervention studies and incorporate objective performance metrics to build on these findings.

Ultimately, this research redefines professional self-concept as both a psychological and cultural resource for nurses. By demonstrating how personal and environmental factors interact to shape clinical decisions, it provides a practical blueprint for developing nursing excellence in Palestine and similar contexts worldwide. The findings bridge theory and practice while affirming the resilience of nurses working under extraordinary pressures.

## Acknowledgments

The authors would like to express their thanks to all the nurses who participated in the study.

## Author contributions

**Conceptualization:** Wasan Aboalrob, Ibrahim Aqtam.

**Data curation:** Wasan Aboalrob.

**Formal analysis:** Ahmad Ayed.

**Investigation:** Wasan Aboalrob.

**Methodology:** Ahmad Ayed, Ibrahim Aqtam.

**Project administration:** Malakeh Z. Malak.

**Resources:** Malakeh Z. Malak.

**Supervision:** Malakeh Z. Malak, Ibrahim Aqtam.

**Validation:** Ahmad Ayed.

**Writing – original draft:** Wasan Aboalrob.

**Writing – review & editing:** Ahmad Ayed, Malakeh Z. Malak, Ibrahim Aqtam.

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
