## [Decision Letter · Decision Letter 0]

27 Jun 2025

Dear Dr. Aqtam,

Thank you for submitting your manuscript to PLOS ONE. After careful consideration, we feel that it has merit but does not fully meet PLOS ONE’s publication criteria as it currently stands. Therefore, we invite you to submit a revised version of the manuscript that addresses the points raised during the review process.

After careful consideration of the reviewer comments, I recommend that the manuscript undergo **major revision** prior to further consideration for publication. While the topic is timely and relevant, reviewers raised concerns regarding the conceptual framework, methodological rigor (particularly sampling and instrument validation), and overinterpretation of correlational findings.

I invite you to revise the manuscript thoroughly, addressing each of the reviewers’ points in a detailed response letter and incorporating the necessary changes into your revised submission.

We look forward to receiving your revised manuscript.

Kind regards,

Nadia Rehman, Ph.D.

Academic Editor

PLOS ONE

Journal Requirements:

2.  We notice that your supplementary figures are uploaded with the file type 'Other'. Please amend the file type to 'Supporting Information'. Please ensure that each Supporting Information file has a legend listed in the manuscript after the references list.

**Additional Editor Comments:**

Q1: While the study is relevant to nursing research, its conceptual contribution is limited. The relationship between self-concept and clinical decision-making is already established in previous literature, including studies cited by the authors themselves. The study replicates prior findings in a different context without sufficient innovation in theoretical framing, measurement, or analysis. Convenience sampling weakens generalizability, and the regression model only explains a moderate variance. A longitudinal or experimental design would have provided more impactful insights.

Q2: The title accurately reflects the content. Abstract is structurally sound but descriptive rather than analytical. It fails to emphasize theoretical novelty or methodological rigor. Recommendations are generic and lack depth.

Q3: The introduction is too descriptive. The study lacks a clear conceptual framework that integrates self-concept theory into clinical decision-making models. Existing theories are not critically evaluated. No justification is offered for the hypothesized direction of influence or the exclusion of mediating/moderating variables such as institutional culture or stress.

Q4: My major concerns. Convenience sampling introduces bias. No justification for using only self-report instruments despite social desirability concerns. No confirmatory factor analysis (CFA) conducted to validate the instruments in the Palestinian context. Regression diagnostics are mentioned but not reported. Use of only cross-sectional correlation limits causal inference.

Q5: Results are clearly presented but lack depth. Key figures such as effect sizes, adjusted R², and confidence intervals are reported but not interpreted.

Q6: Discussion reiterates known associations. The authors overstate implications based on correlational data. The conclusion does not acknowledge measurement limitations, nor does it engage critically with contrasting findings from prior studies. Statements about professional development are not empirically grounded.

Reviewers' comments:

Reviewer's Responses to Questions

**Comments to the Author**

1. Is the manuscript technically sound, and do the data support the conclusions?

Reviewer #1: Partly

Reviewer #2: Yes

Reviewer #3: Yes

2. Has the statistical analysis been performed appropriately and rigorously?

Reviewer #1: N/A

Reviewer #2: Yes

Reviewer #3: Yes

3. Have the authors made all data underlying the findings in their manuscript fully available?

Reviewer #1: Yes

Reviewer #2: Yes

Reviewer #3: Yes

4. Is the manuscript presented in an intelligible fashion and written in standard English?

Reviewer #1: Yes

Reviewer #2: Yes

Reviewer #3: Yes

Reviewer #1: 1. The Clinical Decision-Making in Nursing Scale (CDMNS) and the Nurse Self-Concept Questionnaire (NSCQ) were both completed by the nurses surveyed. Is there a common method bias issue?

2. Structural equation modeling is commonly employed to examine the relationships among multidimensional variables. Are the results obtained through various methods robust?

3. When constructing the regression model, the authors appear to lack a theoretical foundation and seem to select variables solely from a correlational perspective. Have they considered incorporating significant variables mentioned in previous literature, such as gender and work environment stress? For instance, nurses in different types of hospitals may encounter varying environmental stresses. How would the inclusion of these factors affect the relationship between self-concept and clinical decision-making?

4.This study does not further explore the impact of various dimensions of self-concept on clinical decision-making or the underlying mechanisms, which could enhance the practical value of the research. Furthermore, based solely on the results presented in Table 4, the discussion seems to lack depth.

Reviewer #2: This study tackles a significant and underexplored topic the influence of nurses’ professional self-concept on their clinical decision-making (CDM) within the unique context of Palestinian governmental hospitals. The focus on regional healthcare challenges adds valuable insight, and the use of validated instruments such as the Clinical Decision Making in Nursing Scale (CDMNS) and Nurses’ Self-Concept Questionnaire (NSCQ) strengthens the study’s methodological rigor. The sizable sample and multi-site data collection further enhance the robustness of the findings. The manuscript has strong potential to contribute to nursing education and healthcare practice in Palestine and similar contexts. Need some revision before publication.

1. The abstract and introduction lack precise definitions and theoretical grounding for critical constructs such as “self-concept” and “clinical decision-making.” These terms should be clearly defined early on, drawing on established nursing and psychological literature. Clarify self-concept as nurses’ perception of their professional identity and competencies, and explain CDM as a complex cognitive process involving critical thinking and judgment that impacts patient safety.

2. In the introduction, avoid repetition and overly long paragraphs by breaking the content into sub-sections such as “Definition of Self-Concept,” “Impact of Self-Concept on CDM,” and “Context of Nursing in Palestine.” End the introduction with a precise statement of research questions or hypotheses, and explicitly summarize the gap in existing literature that your study addresses.

3. The use of convenience sampling is understandable given the logistical constraints in the Palestinian healthcare context; however, the manuscript should more thoroughly justify this approach and explicitly discuss its limitations on generalizability. Consider discussing potential selection biases and how they may affect findings.

It would be beneficial to propose future research employing random or stratified sampling to enhance external validity. Also, describe strategies implemented (if any) to maximize sample diversity across hospitals, departments, or nurse demographics.

4. The manuscript mentions use of the CDMNS and NSCQ but does not report their reliability statistics within this study’s context. Please provide Cronbach’s alpha coefficients for these scales based on your sample to demonstrate internal consistency.

Discuss whether the instruments have been previously validated in similar cultural or regional contexts, and if any adaptations were made. This strengthens the methodological transparency and trustworthiness of the findings.

5. Statistical findings are reported without sufficient practical interpretation. For instance, what does a mean self-concept score of 205.5 or CDM score of 152.1 indicate in real-world nursing practice? Clarify the scale ranges and what constitutes low, moderate, or high levels.

Additionally, clarify whether assumptions for regression analyses (normality, multicollinearity, homoscedasticity) were tested and met. Reporting these details will strengthen the credibility of your conclusions regarding predictors of CDM.

6. Address discrepancies between text and tables (differing age means) to ensure numerical consistency. Several sentences contain grammatical errors or awkward phrasing that impede readability. For example, revise “decision making judgment” to “decision-making judgment.”

Improve flow by enhancing transitions between paragraphs and avoid repetitive expressions. A language edit focused on conciseness and academic tone, particularly in the discussion and conclusion sections, is strongly recommended.

Reviewer #3: The manuscript titled “Understanding the Influence of Self-Concept on Clinical Decision-Making among Nurses: A Cross Sectional Study” presents a timely and relevant inquiry into the psychological and professional factors that impact clinical decision-making (CDM) in nursing. The study is grounded in validated instruments (CDMNS and NSCQ) and contributes meaningful data from a Palestinian context, which remains underrepresented in international nursing literature. However, while the topic and data are commendable, the manuscript requires major revisions to enhance its academic quality, coherence, and contribution to the field.

1. Clarity and Language Polishing

The overall writing style lacks fluency and consistency in academic tone. Sentences are often verbose and lack cohesion. The authors should revise the manuscript for grammatical accuracy, syntactical clarity, and professional tone. A professional language editor is strongly recommended to enhance readability.

2. Literature Integration and Theoretical Grounding

The literature review is adequate but not analytically deep. The manuscript should more clearly articulate the conceptual framework linking self-concept to CDM. The inclusion of classical theories (Bandura’s self-efficacy, Benner’s novice-to-expert) could strengthen theoretical underpinnings. Additionally, recent studies cited in the discussion (2024 and 2025 references) should be better integrated into the Introduction to frame the research gap earlier.

3. Instrumentation and Validity Reporting

Although the instruments used are reliable, the manuscript does not provide sufficient justification for selecting CDMNS and NSCQ beyond citing Cronbach’s alpha. The authors should explain why these tools were most appropriate for the Palestinian nursing context. Further, the manuscript would benefit from a brief explanation of construct validity, content validation, or cultural adaptation if applicable.

4. Methodological Details

The cross-sectional design is appropriate but limiting. The authors mention a convenience sample yet do not describe sampling strategy, hospital types, or departments in sufficient detail. The inclusion/exclusion criteria should be explicitly listed. Moreover, participant recruitment should be visually summarized (flow diagram).

5. Data Presentation and Interpretation

Tables 1–4 should be clearly labeled with descriptive captions, and all acronyms (M, SD, CDM) must be defined within table footnotes for reader clarity. The regression analysis (Table 4) is under-discussed. The authors should discuss why only self-concept remained significant despite age and experience being correlated in bivariate tests. Further discussion on possible mediation or moderation effects would add value.

6. Discussion and Implications

The discussion repeats many results rather than offering critical insights. The authors should elaborate on the practical implications for nurse managers, educators, and policy-makers. For example, how can nurse training be tailored to enhance self-concept? The impact on patient safety and care quality should be addressed in greater depth.

7. Conclusion and Future Directions

While the conclusion summarizes key points, it would benefit from a more forward looking outlook. The authors should outline specific future research avenues, such as longitudinal designs or intervention-based studies to strengthen causal inferences.

**Do you want your identity to be public for this peer review?** For information about this choice, including consent withdrawal, please see our Privacy Policy

Reviewer #1: No

Reviewer #2: **Yes: ** SUFYAN MAQBOOL

Reviewer #3: **Yes: ** Hafiz Muhammad Ihsan Zafeer

---

## [Author Response · Author response to Decision Letter 1]

28 Jun 2025

Dear Editor and reviewers

Thank you for your time and feedback.

We have addressed all comments in response to reviewers file

Dr Aqtam

---

## [Decision Letter · Decision Letter 1]

17 Jul 2025

Dear Dr. Aqtam,

Thank you for submitting your manuscript to PLOS ONE. After careful consideration, we feel that it has merit but does not fully meet PLOS ONE’s publication criteria as it currently stands. Therefore, we invite you to submit a revised version of the manuscript that addresses the points raised during the review process.

In view of the reviewers' comments, I would like to invite you to address the feedback provided. Please consider revising your manuscript to incorporate the reviewers' suggestions, which will help improve the clarity and quality of your work.

We look forward to receiving your revised manuscript.

Kind regards,

Nadia Rehman, Ph.D.

Academic Editor

PLOS ONE

Journal Requirements:

Reviewers' comments:

Reviewer's Responses to Questions

**Comments to the Author**

Reviewer #1: (No Response)

Reviewer #2: All comments have been addressed

Reviewer #3: All comments have been addressed

2. Is the manuscript technically sound, and do the data support the conclusions?

Reviewer #1: Partly

Reviewer #2: (No Response)

Reviewer #3: Yes

3. Has the statistical analysis been performed appropriately and rigorously?

Reviewer #1: No

Reviewer #2: (No Response)

Reviewer #3: Yes

4. Have the authors made all data underlying the findings in their manuscript fully available?

Reviewer #1: Yes

Reviewer #2: (No Response)

Reviewer #3: Yes

5. Is the manuscript presented in an intelligible fashion and written in standard English?

Reviewer #1: Yes

Reviewer #2: (No Response)

Reviewer #3: Yes

Reviewer #1: 1. The introduction should provide a more detailed explanation of why this issue is being studied and the significance of the research. It is necessary to add a solid empirical foundation. Currently, the literature review is insufficient, and incorporating studies from various countries would enhance its depth. Additionally, a conceptual definition should be included, highlighting the perspectives of different scholars. This comparison of how various researchers define key concepts, along with an exploration of measurement methods and authoritative measurement tools, will contribute to a richer research content.

2. The formulation of the research hypotheses appears to be inadequately supported by a theoretical basis.

3. The conclusions are somewhat superficial, and the academic rigor is insufficiently represented. The methodology is relatively simplistic, and the conclusions seemingly lack credibility.

4. The overall logic of the paper is relatively weak, and the focus of the content is not clearly defined. It is recommended to organize the material into 3-4 sections based on the study's logical framework (e.g., introduction, literature review, research methodology, analysis of results, discussion, and conclusions). Additionally, the study seemingly lacks depth.

Reviewer #2: (No Response)

Reviewer #3: The authors have adequately addressed the reviewer’s comments and made the necessary revisions to strengthen the manuscript. The responses are thorough and demonstrate a clear effort to enhance the clarity, methodological rigor, and overall quality of the study. Therefore, I recommend the manuscript for acceptance.

**Do you want your identity to be public for this peer review?** For information about this choice, including consent withdrawal, please see our Privacy Policy

Reviewer #1: No

Reviewer #2: **Yes: ** Sufyan Maqbool

Reviewer #3: No

---

## [Author Response · Author response to Decision Letter 2]

18 Jul 2025

Dear Editor’s an Reviewers,

On behalf of all co-authors, I would like to express our sincere gratitude to you and the reviewers for the time, effort, and insightful feedback provided on our manuscript titled "Understanding the Influence of Self-Concept on Clinical Decision-Making among Nurses: A Cross-Sectional Study."

We greatly appreciate the constructive comments and suggestions, which have helped us to improve the clarity, quality, and rigor of our work. In response, we have carefully addressed all the points raised and provided detailed, point-by-point responses to each comment in the “Response to Reviewers” file.

Dr Aqtam

---

## [Decision Letter · Decision Letter 2]

8 Aug 2025

Understanding the Influence of Self-Concept on Clinical Decision-Making among Nurses: A Cross-Sectional Study

PONE-D-25-29031R2

Dear Dr. Aqtam,

We’re pleased to inform you that your manuscript has been judged scientifically suitable for publication and will be formally accepted for publication once it meets all outstanding technical requirements.

Kind regards,

Nadia Rehman, Ph.D.

Academic Editor

PLOS ONE

Additional Editor Comments (optional):

Reviewers' comments:

Reviewer's Responses to Questions

**Comments to the Author**

Reviewer #1: All comments have been addressed

2. Is the manuscript technically sound, and do the data support the conclusions?

Reviewer #1: Yes

3. Has the statistical analysis been performed appropriately and rigorously?

Reviewer #1: Yes

4. Have the authors made all data underlying the findings in their manuscript fully available?

Reviewer #1: Yes

5. Is the manuscript presented in an intelligible fashion and written in standard English?

Reviewer #1: Yes

Reviewer #1: The authors have provided all additional responses to the previous questions. It is recommended to accept this manuscript.

**Do you want your identity to be public for this peer review?** For information about this choice, including consent withdrawal, please see our Privacy Policy

Reviewer #1: No

---

## [Editor Report · Acceptance letter]

PONE-D-25-29031R2

PLOS ONE

Dear Dr. Aqtam,

I'm pleased to inform you that your manuscript has been deemed suitable for publication in PLOS ONE. Congratulations! Your manuscript is now being handed over to our production team.

Kind regards,

on behalf of

Dr. Nadia Rehman

Academic Editor

PLOS ONE